# Low-Dose Radiation Therapy for COVID-19: A Systematic Review

**Seyed Mohammad Javad Mortazavi** [1] , **Seyedeh Fatemeh Shams** [1], **Sahar Mohammadi** [2],
**Seyed ALi Reza Mortazavi** [3] **and Lembit Sihver** [4,5,*]

[1]  Medical Physics and Engineering Department, School of Medicine, Shiraz University of Medical Sciences, Shiraz 71348-45794, Iran; mortazavismj@gmail.com (S.M.J.M.); f.shams7242@yahoo.com (S.F.S.)

[2]  Department of Radiologic Technology, Behbahan Faculty of Medical Sciences, Behbahan 63617-96819, Iran; saharmohammadi44487@gmail.com

[3]  School of Medicine, Shiraz University of Medical Sciences, Shiraz 71348-45794, Iran; alireza.mortazavi.med@gmail.com

[4]  Department of Radiation Physics, Atominstitut, Technische Universität Wien, Stadionallee 2, 1020 Vienna, Austria

[5]  Department of Physics, Chalmers University of Technology, 412 96 Gothenburg, Sweden

*  Correspondence: lembit.sihver@tuwien.ac.at; Tel.: +46-721-726431

**Simple Summary:** There are limited available data indicating that in oxygen-dependent elderly patients with COVID-19-associated pneumonia, low-dose whole-lung radiation doses, ranging from 0.5 to 1.5 Gy, can lead to accelerated recovery and progress in clinical status, encephalopathy, and radiographic consolidation without any detectable acute toxicity. Therefore, low-dose radiation therapy (LDRT), using conventional cancer radiation therapy machines, could be introduced as a safe treatment with promising efficacy that fully warrants further large-scale studies. Current findings indicate that LDRT could increase the survival of elderly patients and of patients with genetic risk factors, who are at greater risk of mortality due to COVID-19, even if more preclinical work and clinical trials are needed before any clear conclusion can be made.

**Abstract:** The ongoing COVID-19 pandemic is of great concern for the whole world, and finding an effective treatment for the disease caused by the severe acute respiratory syndrome coronavirus-2 (SARS-CoV-2) is, therefore, a global race. In particular, treatment options for elderly patients and patients with genetic risk factors with COVID-19-associated pneumonia are limited, and many patients die. Low-dose radiotherapy (LDRT) of lungs was used to treat pneumonia many decades ago. Since the first report on the potential efficacy of LDRT for COVID-19-associated pneumonia was published on 1 April, 2020, tens of papers have addressed the importance of this treatment. Moreover, the findings of less than 10 clinical trials conducted to date are now available. We performed a detailed search of PubMed/MEDLINE, Web of Science, Google Scholar, and Scopus and selected the nine most relevant articles. A review of these articles was conducted. The available data indicate that in oxygen-dependent elderly patients with COVID-19-associated pneumonia, whole-lung radiation at doses of 0.5–1.5 Gy can lead to accelerated recovery and progress in clinical status, encephalopathy, and radiographic consolidation without any detectable acute toxicity. Although data collected so far show that LDRT could be introduced as a treatment with promising efficacy, due to limitations such as lack of randomization in most studies, we need further large-scale randomized studies, especially for elderly patients who are at greater risk of mortality due to COVID-19. However, more preclinical work and clinical trials are needed before any clear conclusion can be made.

**Keywords:** SARS-CoV-2; COVID-19; low-dose radiation; radiotherapy; selective pressure

## 1. Introduction

COVID-19 is a respiratory disease caused by the single-stranded RNA virus SARS-CoV-2 [1–5]. This disease was first observed in the city of Wuhan, the capital of the Hubei province in China, in December 2019. Since December 31, 2019, until the end of March

in 2021, 129 million cases of COVID-19, including 2.8 million deaths (in accordance with the applied case definitions and testing strategies in the affected countries), have been reported [6]. A large fraction of COVID-19 patients appear to be asymptomatic, and many other patients may only experience mild symptoms such as fever, cough, anosmia, and myalgia [7–9]. However, a subset of patients develops high-grade fever, cough, and dyspnea.

As addressed in the early report of Ghadimi-Moghadam et al. [10], using X-ray therapy to treat pneumonia dates back to the first half of the twentieth century in the preantibiotic era, but the efficacy of this treatment approach is unsure. A review by Calabrese and Dhavan, published 2013 [11], presented 15 reports covering 863 patients with severe pneumonia of different pathogeneses. Two of the studies treated patients with low doses of kilovoltage X-rays, and the reported clinical responses were good with a reduction in mortalities. In 1943, Oppenheimer reported the outcome of 56 patients with presumed viral pneumonia treated with 0.35–0.9 Gy using 130–150 kVp X-rays [12]. However, as also discussed in [13], Oppenheimer concluded that roentgen therapy of pneumonia virus was useful mainly during the early stages of the disease, and he could not prove that the 56 patients of his series would not have recovered as rapidly even without roentgen therapy [12]. In the same year, Correll and Cowan reported a second case series of 155 patients with viral pneumonia [14]. These patients had fever, sore throat, and chills but no dyspnea. Most of the patients received supportive care alone or were treated with antibiotics. The average duration of illness was approximately 12 days. A subset of 23 patients received 1.12 Gy with 100 kVp X-rays to the involved lobe of the lung, which was repeated 24 h later in most patients because there was no satisfactory clinical response. Low-dose radiotherapy (LDRT) is a technique that historically was used for many noncancer pathologies such as arthrodegenerative and inflammatory diseases. [15]. The two case series by Oppenheimer and Correll and Cowan do not provide any strong evidence that LDRT can cure acute respiratory distress syndrome (ARDS) caused by the SARS-CoV-2 virus, but with large numbers of patients dying from COVID-19 and because there are no currently approved treatments of patients with the virus, some researchers have proposed testing low-dose ($\leq 1$ Gy) radiotherapy to the thorax for COVID-19 pneumonia [16]. This therapeutic approach in the early trials of treating pneumonia with X-rays [12] was based on an incomplete understanding of the anti-inflammatory effects of low-dose radiation (LDR) [11,17]. It is worth noting that the term "low-dose radiation," as defined by UNSCEAR, comprises doses below 100 mGy. However, in this review, "low-dose radiation" does not apply to the radiation protection context and is only used for the specific domain of the medical therapeutic application of radiation.

It is known that the relationship between irradiation and inflammatory response is strongly dependent on dose and dose rate. While higher levels of ionizing radiation lead to proinflammatory responses, LDR exposure can activate anti-inflammatory molecules such as TGF-b1 and IL-10 [18,19]. These anti-inflammatory effects following LDRT have been known and utilized for decades [3,15,18–21]. Furthermore, as oxidative stress plays a key role in thrombosis through mechanisms such as increased production of isoprostanes [22], LDRT might decrease or prevent thrombosis through reduction of the oxidative stress level [23]. The main goals of using LDRT for treatment of COVID-19-associated symptoms are summarized in Table 1.

**Table 1.** Cardinal goals of using LDRT for treatment of COVID-19-associated symptoms.

| Management of Pneumonia, ARDS, and Other Fatal Changes Associated with COVID-19 | |
| --- | --- |
| **Low-Dose Radiation (LDR) Triggers:** | **Low-Dose Radiation (LDR) Inhibits:** |
| ○ Anti-inflammatory effects | ○ Cytokine-releasing cells |
| ○ Antithrombosis effects | ○ Selective pressure |
| ○ Immune system optimization and metabolic rewiring | ○ Adaptive mutations and viral evolution |
| ○ Alveolar acceleration<br>○ Mucus absorption | ○ Emergence of new variants with more virulence and transmissibility |

This systematic review is an attempt to answer the question of whether LDRT is an effective method for treating oxygen-dependent patients with COVID-19-associated pneumonia. Recent studies of LDRT on mice with moderate lung injury induced by bleomycin identified 1.0 Gy as the most effective radiation dose tested and revealed plausible immunological mechanisms that support the notion that LDRT is worth investigating as a treatment of COVID-19-associated pneumonia [24]. Papachristofilou et al. [25] studied the use of whole-lung low LDRT in twenty-two patients admitted to an intensive care unit (ICU) who required mechanical ventilation for COVID-19 pneumonia. The patients, who were generally elderly and comorbid with a median age of 75 years, were randomized to either whole-lung LDRT or sham-RT between November and December 2020. The results from this study showed that whole-lung LDRT failed to improve clinical outcomes in critically ill patients requiring mechanical ventilation for COVID-19 pneumonia. The in vitro study conducted by Meziani et al. [26] was removed due to its structure. However, a more recent study on 36 patients who received a single thoracic dose of 0.5 Gy confirmed the efficacy of LDRT for pneumonia in COVID-19 patients [27].

## 2. Materials and Methods

### 2.1. Literature Search

The present study was conducted as a systematic review of publications discussing LDRT for treating COVID-19 patients. PRISMA guidelines were used in this systematic review. Searching for articles published from inception to the end of March 2021 was performed in the PubMed/MEDLINE, Web of Science, Google Scholar, Scopus, and ScienceDirect databases using the terms "low dose radiation therapy" "LDRT," "pneumonia," "COVID-19," "SARS-CoV-2," and "whole lung irradiation. After selecting the appropriate articles and rejecting the irrelevant ones, they were encoded, and analysis of the findings and discussions was carried out.

### 2.2. Inclusion and Exclusion Criteria

The inclusion criteria for our systematic review include: (1) All published original articles reporting on the therapeutic effects of LDRT for COVID-19, (2) written in English, (3) published between January 2020 and July 2021, (4) sample size >2 patients. Exclusion criteria: editorials, letter to the editor (LTE) articles without actual data, commentaries, and case series, and case reports were excluded.

After removing duplicates, we analyzed the titles and abstracts of the remaining articles independently to assess if they meet our inclusion criteria. Overall, 710 articles were in the initial search, and 65 articles were about using LDRT for treating pneumonia in COVID-19 patients. By reviewing the abstracts and removing letters, reviews, or hypotheses, 10 papers that were focused on LDRT for COVID-19 were selected. Moreover, a Monte Carlo numerical simulation study on the interaction of an electron beam with the novel coronavirus was also removed from the present study. Finally, 9 clinical trial studies remained for the review (Figure 1). Given this consideration, the effect of whole-lung low-dose radiotherapy in COVID-19 patients was investigated in these clinical trials.

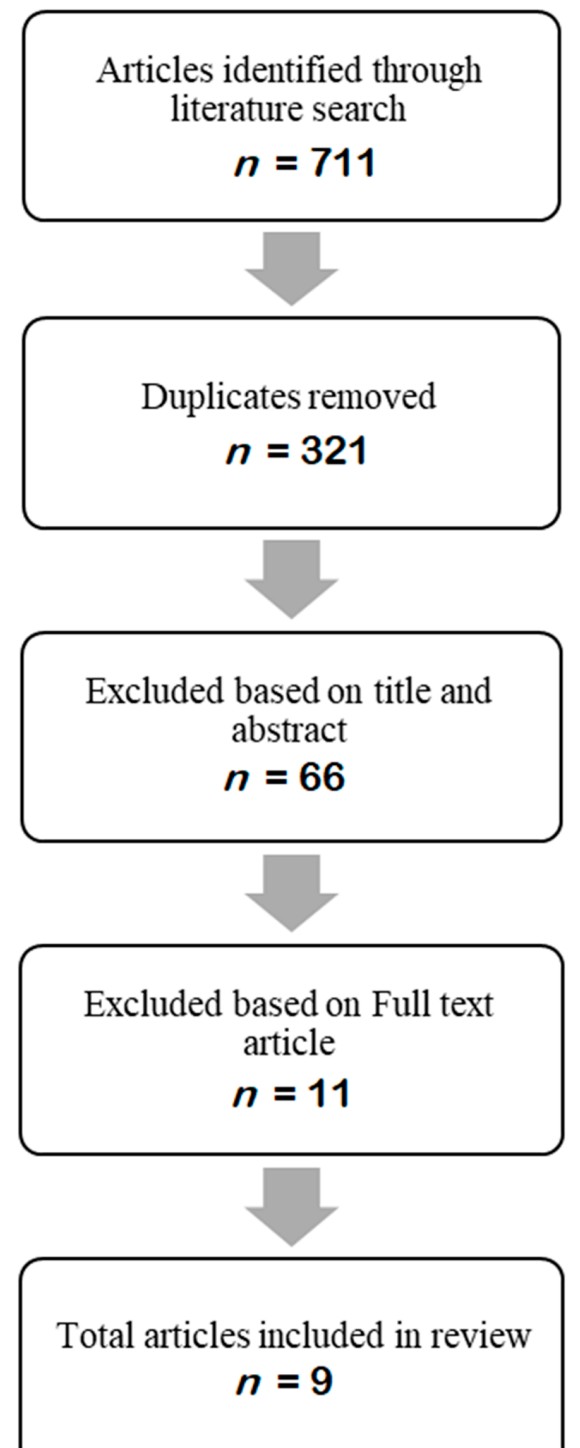

**Figure 1.** Flowchart of studies included in this systematic review.

In order to perform the systematic review of the studies, the articles were classified based on the month of publication, sample size, method of research, clinical outcomes, and final results. The selected studies included data on using LDRT for COVID-19-associated pneumonia, hospitalization, radiographic findings (e.g., consolidation), need for supplemental oxygen, and clinical evaluation of the illness. The articles were categorized based on the month of publication. Most studies were conducted in November 2020.

*2.3. Sample Size*

Nine articles described investigations into the effect of LDRT on patients, while Meziani et al. [26] presented an in vitro study. In Figure 2, the sample sizes of the 9 clinical trials reviewed in this study are shown. The highest number of samples in any of the trials was 36 patients (Arenas et al. [27]), while the lowest number of samples was only 2 (Moreno et al. [28]).

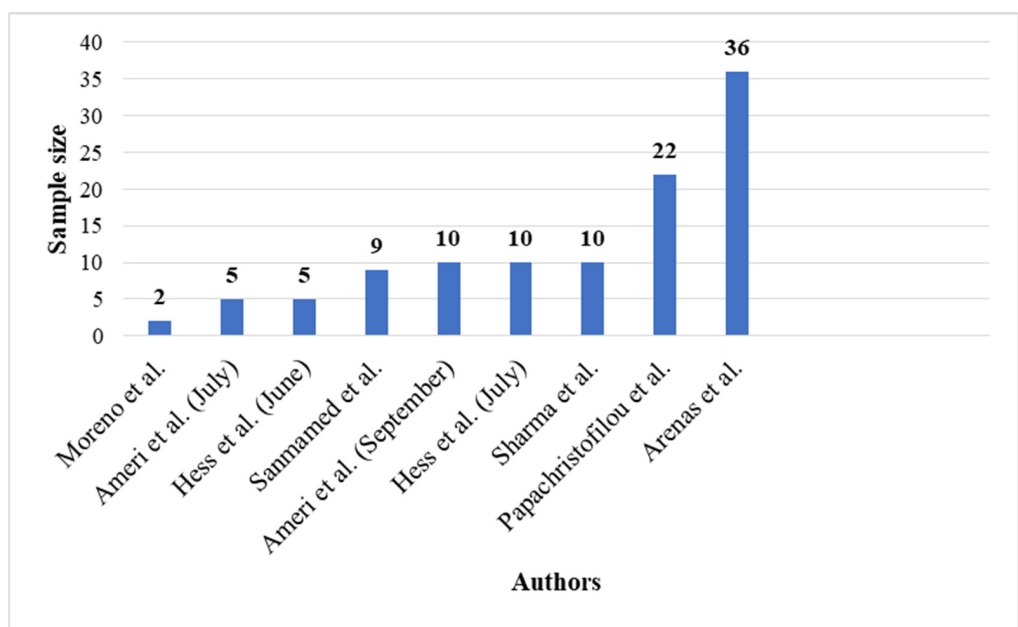

**Figure 2.** Sample size in the 9 investigated studies.

*2.4. Low-Dose Radiation Therapy*

In the 9 clinical trials finally reviewed in detail in this study, patients received a single fraction of radiation to the lungs at doses between 0.5 and 1.5 Gy. The radiation dose was delivered via an anterior–posterior beam configuration in all these trials [26–35]). The highest dose was 1.5 Gy [32,33], and the lowest dose was 0.5 Gy [26,29,30]. The variety of radiation doses are summarized in Figure 3.

*2.5. Studied Parameters*

The different measured parameters studied in the different studies included in our review are tabulated in Table A1 in Appendix A. As outlined in Table A1, C-reactive peptide (CRP), D-dimer, IL-6, and ferritin were investigated in all 10 studies, excluding the in vitro study conducted by Meziani et al. Among the reviewed studies, Hess et al. [32] studied the highest number of parameters (15 parameters).

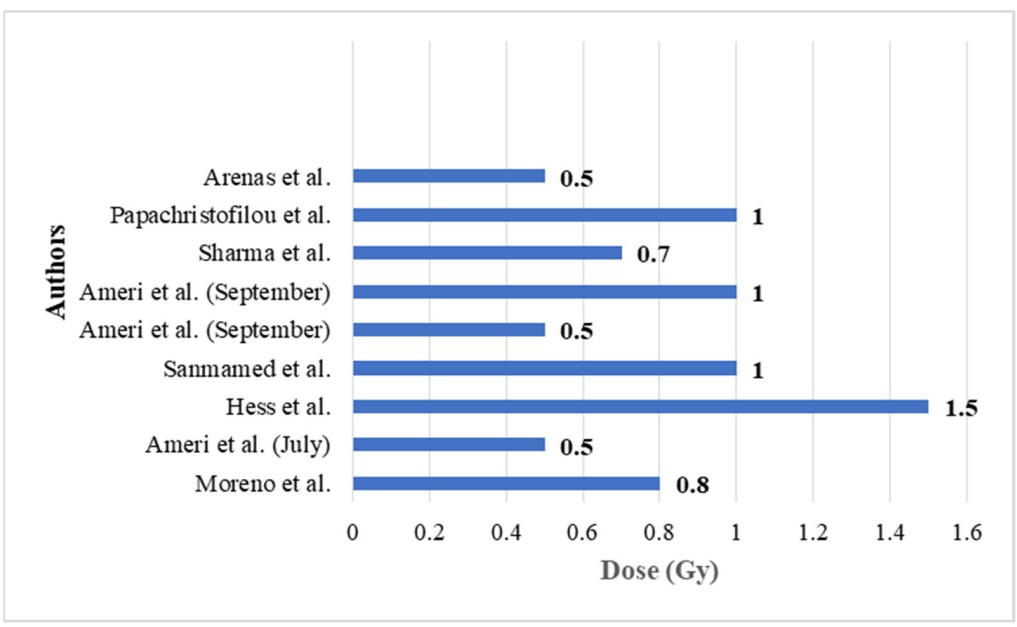

**Figure 3.** Radiation dose in the 9 clinical trials included in this review.

### 3. Results

The results from the studies included in our review are summarized in Table A2 in Appendix A. All studies introduced whole-lung LDRT as a promising approach for avoiding or delaying invasive respiratory support. In all studies except one, more than 60% of patients showed improvement in their clinical and radiological findings and survival. However, before larger clinical trials should be considered, further preclinical work is needed to demonstrate the efficacy of LDRT in avoiding or delaying invasive respiratory support. In one trial conducted in Switzerland, LDRT was reported to show no improvement in clinical outcomes in critically ill patients who needed mechanical ventilation in COVID-19 patients with pneumonia [25].

Arenas et al. recently conducted the largest clinical trial so far. Due to its crucial importance, we will explore this study and its findings in detail. Arenas et al. evaluated the efficacy of low-dose radiation therapy (LDRT) for pneumonia in COVID-19 patients [27]. In their study, 36 patients received a single thoracic dose of 0.5 Gy. All patients received dexamethasone. While 13 of 36 patients died within 1–25 days after radiation therapy, the remaining 23 patients (64%) showed improvement one week after LDRT. Despite some strength, the paper authored by Arenas et al. has a number of shortcomings. Firstly, we suspect that the use of dexamethasone was one factor that negatively affected the outcome of the trial performed by Arenas et al. The indiscriminate use of dexamethasone may have weakened the patients' immune systems, resulting in more rapid replication of the SARS-CoV-2. In this trial, 13 patients died after LDRT (eight died of COVID-19 disease and five of other causes such as esophageal variceal hemorrhage, pulmonary thromboembolism, septic infection, bronchoaspiration, and chronic renal failure). Despite reports on the therapeutic advantage of dexamethasone, this benefit was limited to seriously ill COVID-19 patients. Patients with milder disease did not derive any advantage. The basic characteristics of the trials completed to date are summarized in Table 1. The studies summarized in the table were performed using radiation doses ranging between 0.5 and 1.5 Gy for patients in different age ranges. Another potential shortcoming associated with the use of steroids (e.g., dexamethasone) is related to suboptimal immunity and the selective pressure induced by these drugs driving the virus to mutations. Moreover, systemic corticosteroid therapy has been introduced as a double-edged sword in viral pneumonia due to its role in immunosuppression and increasing the risk of mucormycosis in susceptible individuals [36].

It should be noted that while optimal immunity limits viral replication and hence limits mutations and the likelihood of the emergence of new variants, any factor that causes suboptimal immunity may result in more rapid replications of the virus and a higher likelihood of the emergence of new variants [37]. Viral mutations may generate new variants with altered virus stability, transmissibility, virulence, and pathogenesis. Moreover, based on ongoing experience in India, dexamethasone is possibly linked to an increased incidence of potentially fatal mucormycosis among diabetic COVID patients. The third shortcoming of this study comes from the very limited age range of the patients who participated in this study (84 ± 8.1 years). This restricted and elderly age range makes the patients more prone to common diseases found in older populations (e.g., cardiovascular diseases, influenza, and cellular immune dysfunction). In addition, the study does not provide any information about the therapeutic efficacy of LDRT in other age ranges. Although the study by Arenas et al. provides additional data regarding pulmonary low-dose radiation therapy, it must be viewed in terms of its limited scope.

## 4. Discussion

LDRT might mitigate COVID-19 pneumonitis by inducing an anti-inflammatory effect. Many animal and human in vivo studies, as well as in vitro studies, have shown that LDRT can control bacterial pneumonia [38]. It is well known that X-ray/γ-ray radiotherapy is a cost-effective cancer treatment that is easily available in most hospitals [39], thus Linacs are available in many hospitals around the world, providing great strength for LDRT for COVID-19. Our review gives further support to the feasibility of utilizing LDRT for COVID-19 patients with moderate to severe illness [35]. Despite its great advantages, there are concerns about the possible risks, including carcinogenesis, cardiovascular, and spinal cord damage of radiation exposure to patients with COVID-19. Furthermore, delivering radiation therapy to hypoxic COVID-19 patients, who may have ARDS, could increase the risk of infection to staff and other patients if not enough safety measures are taken. However, current data indicate that the cancer risk associated with LDRT is scarce [28,40,41]. LDRT can affect the lung macrophages significantly at 0.5 and 1 Gy radiation doses. Exposure of human lung macrophages to LDR may decrease IFNγ production and increase IL-10 secretion. In addition, LDRT can increase the percentage of human lung macrophages that produce IL-10. However, LDRT decreases the percentage of human lung macrophages that produce IL-6 [26]. Moreover, compared to antiviral therapy, LDRT might prevent selective pressure-induced adaptive mutations [10,38].

We believe that the lack of a good systematic review has led to unjustified exposure of COVID-19 patients to high doses of radiation in different trials around the world. A recent randomized trial on 22 elderly COVID patients indicated that whole-lung LDRT could not improve clinical outcomes in critically ill COVID-19 patients with pneumonia who needed mechanical ventilation [25]. In this trial, the patients were randomly exposed to either 1 Gy dose to the whole lung region or a sham irradiation. To better explore the origin of such a failure, it is worth noticing that substantial evidence shows that 1 Gy may be far beyond the range of therapeutic radiation doses. In March 2020, Ghadimi-Moghadam first proposed LDRT for COVID-19-associated pneumonia using doses up to 250 mGy [10]. However, later, different researchers around the world, in competition, increased the radiation doses. Emory University Hospital used 1.5 Gy, and Ameri et al. tried both 0.5 Gy and 1.0 Gy. Interestingly, a paper published recently in Environment International [42] clearly indicates that while doses <1 Gy have anti-inflammatory effects, doses >1 Gy have proinflammatory effects and cause fibrosis, as illustrated in Figure 4. Thus, the doses used in the clinical trials such as the study conducted by Emory University Hospital were possibly unjustified. Moreover, using a relatively high dose might be the reason that, in Switzerland, researchers failed to show any therapeutic effects for LDRT [38]. In addition, this point clearly clarifies why Ameri et al. finally confessed that, in their trial, 0.5 Gy was more effective than 1.0 Gy [29].

It should be noted that moving to doses less than 1 Gy (preferably <0.5) not only improves the therapeutic effects of LDRT but also decreases the cancer risk to an acceptable level. Arruda et al. recently reported that enrolling patients aged >40 years and, in particular, elderly patients of >60 years of age, regardless of their sex, can provide an acceptable lifetime attributable risk (LAR) of radiation-induced cancer (RIC) for a radiation dose of 0.7 Gy [43]. They stated that only 0.5 Gy had an acceptable risk of exposure-induced death (REID). These researchers concluded, "The current ongoing trials should initially use doses ≤ 0.5 Gy to maintain the risks at an acceptable level and include only patients who fail or do not have any other treatment option". Although the paper by Arruda et al. is a great contribution in the field of LDRT for COVID-19, it has some omissions that are addressed by Bevelacqua et al. [44]. It is of crucial importance to note that using doses ≤0.5 Gy not only increases the therapeutic effects of LDRT and maintains the risks at an acceptable level but also decreases the cancer risk to a justified level. Studies conducted on acute radiation sickness (ARS) in Chernobyl show that an instant dose in the range of 0.5 to 0.7 Gy or higher may cause significant bone marrow damage [45]. The radiation doses needed for minimal and clinically significant effects in bone marrow compared to doses used by Hess et al. [32], Ameri et al. [29], Papachristofilou et al. [25], and Sanmamed et al. [35], who used doses ≥1 Gy in their trials are indicated in Table 2.

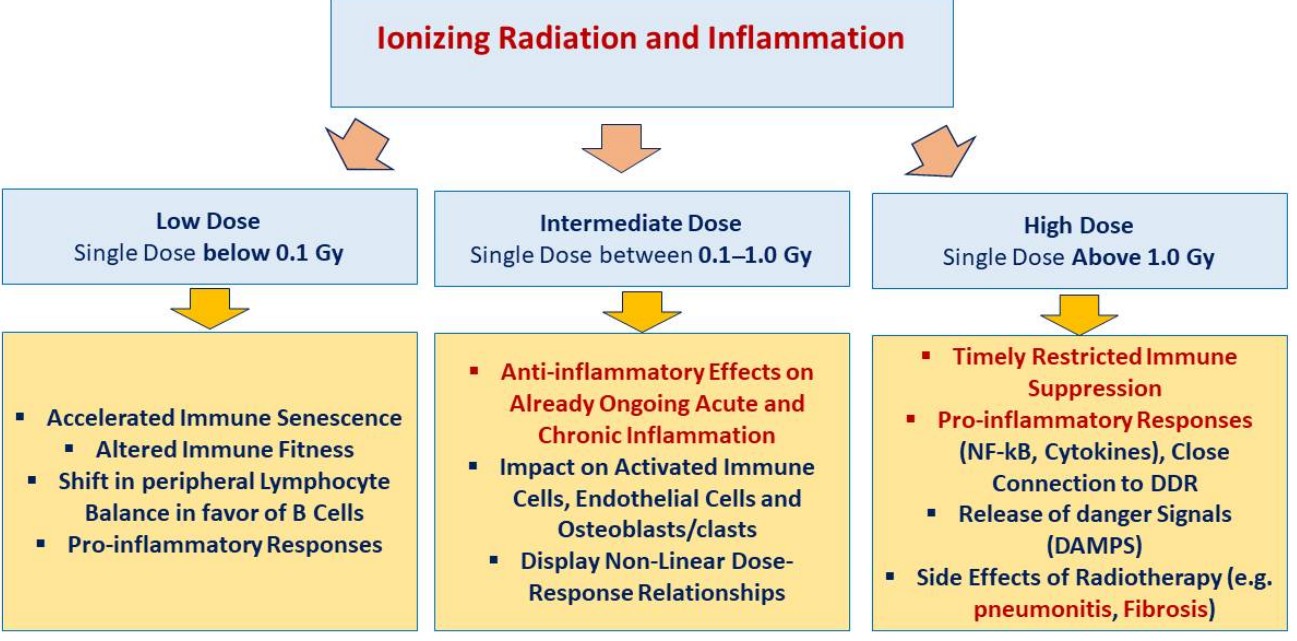

**Figure 4.** Some of the most important immune- and inflammation-related processes that develop after low-, intermediate-, and high-dose irradiation. (Reproduced from [42]).

**Table 2.** The radiation doses required for minimal and clinically significant effects in bone marrow compared to doses used in some clinical trials.

| | |
|---|---|
| Hess et al. | 1.5 Gy |
| Ameri et al. (2nd phase) | 1.0 Gy |
| Papachristofilou et al. | 1.0 Gy |
| Sanmamed et al. | 1.0 Gy |
| Minimal effect in bone marrow | 0.5–0.7 Gy |
| Clinically significant effect in bone marrow | >1 Gy |

## 5. Conclusions

It is known that LDRT might mitigate COVID-19 pneumonitis by inducing an anti-inflammatory effect. A screening of all articles published since 1 April 2020, on the potential efficacy of LDRT of COVID-19 was conducted, and nine clinical trials were finally reviewed in detail. In these trials, patients received a single fraction of radiation, delivered via an anterior–posterior beam configuration, to the lungs at doses between 0.5 to 1.5 Gy. In all studies except one, more than 60% of patients showed improvement in their clinical and radiological findings and survival. The results, therefore, showed that in oxygen-dependent elderly patients with COVID-19-associated pneumonia, whole-lung radiation at doses of 0.5–1.5 Gy can lead to accelerated recovery and progress in clinical status, encephalopathy, and radiographic consolidation without any detectable acute toxicity. None of the studies included in our review showed significant acute radiotoxicity compared to age, sex, and comorbidity-matched controls. Although these data showed that LDRT could be introduced as a treatment with promising efficacy, due to limitations such as lack of randomization in most studies, we need further large-scale randomized studies, especially for elderly patients who are at greater risk of mortality due to COVID-19. Key issues that should be addressed in future studies are finding the optimum radiation dose and dose rate and how to secure the safety of the patients and medical staff during the treatments. Long-term follow-up of patients will provide further information that helps COVID-19 management policy makers better evaluate this therapeutic approach. Considering the small sample size of all studies included in our review, both preclinical work and more clinical studies, with a larger number of patients, are needed to confirm the safety and effectiveness of LDRT for COVID-19-related pneumonia.

**Author Contributions:** S.M.J.M. and L.S. Conceived and designed the analysis. S.F.S., S.M., S.A.R.M., S.M.J.M., L.S. have contributed equally to conducting the literature review and writing and editing the article. S.M.J.M. and L.S. finalized the review. All authors have read and agreed to the published version of the manuscript.

**Funding:** This research received no external funding.

**Institutional Review Board Statement:** Not applicable.

**Informed Consent Statement:** Not applicable.

**Data Availability Statement:** Not applicable.

**Conflicts of Interest:** The authors declare no conflict of interest.

# Appendix A

**Table A1.** Measured parameters in the reviewed studies.

| | Measured Parameters | Ameri et al. (July) [30] | Ameri et al. (September) [29] | Arenas et al. [27] | Hess et al. [33] | Hess et al. [32] | Moreno et al. [28] | Sanmamed et al. [35] | Sharma et al. [35] | Papachristofilou et al. [25] |
|---|---|---|---|---|---|---|---|---|---|---|
| 1 | CRP | * | * | * | | * | * | * | * | * |
| 2 | Lactate dehydrogenase | | | | | * | | | | |
| 3 | Creatine kinase | | | | | * | | | | |
| 4 | D-dimer | | * | * | | * | * | * | * | |
| 5 | troponin | | | | | * | | | | |
| 6 | Aspartate aminotransferase (AST) | | | | | * | | | | |
| 7 | Alanine aminotransferase (ALT) | | | | | * | | | | |
| 8 | White blood cell count | | | | | * | | | | |
| 9 | Creatinine | | | | | * | | | | |
| 10 | Interleukin-6 | * | * | * | | * | * | | * | |
| 11 | Myoglobin | | | | | * | | | | |
| 12 | Fibrinogen | | | | | * | | | | |
| 13 | Erythrocyte sedimentation rate | | | | | * | | | | |
| 14 | Ferritin | | * | * | | * | * | * | * | * |
| 15 | Procalcitonin | | * | | | * | | | | |
| 16 | LDH | | | * | | | * | * | | |
| 17 | Leucocyte | | | * | | | | | | |
| 18 | Glutamate pyruvate transaminase (GPT) | | | | | | | | | |
| 19 | Hemoglobin | | | | | | * | | | |
| 20 | Lymphocyte | | | * | | | * | | | * |
| 21 | Platelet | | | * | | | | | | |
| 22 | Fibrinogen | | | | | | * | | | |
| 23 | SatO$_2$/FiO$_2$ index (SAFI) | | | * | | | | * | | * |
| 24 | SpO$_2$ | * | * | * | * | | | * | | |
| 25 | Temperature | * | * | | | * | | | | |

* parameters measured in each study.

**Table A2.** Summary of the studies.

| Study Author(s) | Start Date | Location | Number of Patients | Mean Age | Interventions | Dose Radiation | Total Oxygen Supplementation Duration | Discharge | Criteria for Efficiency of LDRT | Potential Biases | Outcome |
|---|---|---|---|---|---|---|---|---|---|---|---|
| Ameri et al. | 21 May 2020 and July 2020 | Imam Hossein Hospital, Tehran, Iran | 10 | 75 | (1) Standard national guideline for the management of COVID-19: (1) Supplemental oxygen (preferably) via high-flow nasal cannula, (2) unfractionated heparin 5000 units subcutaneously every 8 h or enoxaparin 40 mg subcutaneously once daily, (3) antibiotics (if clinically indicated; e.g., community-acquired pneumonia), (4) basic supportory care, (5) careful monitoring of patients for clinical indices, and (6) dexamethasone 8 mg daily for up to 10 days (at the physician's discretion) (2) Single-fraction whole-lung radiotherapy | 0.5 or 1 Gy | All patients received $O_2$ supplementation mainly (60%) via facial masks with reservoir bags | Median: 6th day; range: 2nd–14th days | Primary endpoints: improvement in $SpO_2$, the number of hospital/intensive care unit (ICU) stay days, and the number of intubations performed after RT secondary endpoints: changes in laboratory test results (including CRP, IL-6, ferritin, procalcitonin, and D-dimer) following RT | | 0.5 Gy LDRT: rise in $SpO_2$: 80% clinical recovery (included patients who were discharged from the hospital or acquired $SpO_2$ ≥93% on room air): 75% 1 Gy LDRT: rise in $SpO_2$: 40% clinical recovery: 40% |
| Ameri et al. | 21 May 2020 and 24 June 2020 | Imam Hossein Hospital, Tehran, Iran | 5 | 71.8 | Single-fraction whole-lung radiotherapy | 0.5 Gy | Four of the patients recovered rapidly and were weaned from supplemental oxygen at a mean time of 1.5 days | 7 days | Vital signs (including blood oxygenation and body temperature) and laboratory findings (interleukin-6 and C-reactive peptide) | | Clinical and paraclinical findings of 4 of the 5 patients improved on the first day of irradiation |

Table A2. *Cont.*

| Study Author(s) | Start Date | Location | Number of Patients | Mean Age | Interventions | Dose Radiation | Total Oxygen Supplementa-tion Duration | Discharge | Criteria for Efficiency of LDRT | Potential Biases | Outcome |
|---|---|---|---|---|---|---|---|---|---|---|---|
| Arenas et al. | - Between June and November 2020 | Spain | 36 | 84 | Dexamethasone treatment Single-fraction whole-lung radiotherapy | 0.5 Gy | | | Primary endpoints: increasing in the ratio of arterial oxygen partial pressure (PaO$_2$) or the pulse oximetry saturation (SpO$_2$) to fractional inspired oxygen (FiO$_2$) ratio of at least 20% at 24 h with respect to the preirradiation value | | Mean SpO$_2$ pretreatment value was 94.28% and the SpO$_2$/FiO$_2$ ratio varied from 255 mm Hg to 283 mm Hg at 24 h and to 381 mm Hg at 1 week |
| Moreno-Olmedo et al. | April, 2020 | La Milagrosa-Hospital (Madrid, Spain) | 2 | 72.5 | (1) The medical therapy administered to both patients consisted of lopinavir/ritonavir, hydroxychloroquine, azithromycin, piperacillin/tazobactam, prophylactic doses of low-molecular-weight heparins (LMWHs), corticosteroids (methylprednisolone 250 mg × 3 boluses) and tocilizumab (single dose) (2) Single-fraction whole-lung radiotherapy | 0.8 Gy | (1) Patient 1 showed an improvement on his O$_2$-Sat and PaFi02 (>300) two days after the treatment (2) Patient 2 showed a slower recovery, achieving less need for oxygen support 2, 5, and 7 days after the treatment | 8 and 14 days | Primary endpoints: achieving hospital discharge Radiological improvement secondary endpoints: SatO$_2$ | | Radiological improvement, achieving hospital discharge after 1 radiotherapy session over a period of 8 and 14 days SatO$_2$ > 93% |

**Table A2.** *Cont.*

| Study Author(s) | Start Date | Location | Number of Patients | Mean Age | Interventions | Dose Radiation | Total Oxygen Supplementation Duration | Discharge | Criteria for Efficiency of LDRT | Potential Biases | Outcome |
|---|---|---|---|---|---|---|---|---|---|---|---|
| Hess et al. | 23 April to 24 May 2020 | - | 10 | 78 | (1) Patients received best supportive care plus single-fraction whole-lung radiotherapy (2) Patients in the control cohort received best supportive care with or without COVID-directed therapies (i.e., remdesivir, hydroxychloroquine, glucocorticosteroids, etc.) per protocol or physician discretion | 1.5 Gy | Median total time requiring oxygen supplementation was 10 days | Median time to hospital discharge: 20 and 12 days | Efficacy endpoints: time to clinical recovery, radiographic improvement, and serologic responses | | Clinical recovery: 3 days for LDRT Median time to hospital discharge: 12 days, intubation rates: 10%, The LDRT cohort had faster radiographic improvement |
| Hess et al. | 24 and 28 April 2020 | Emory University, Atlanta, U.S. | 5 | 90 | (1) Single-fraction whole-lung radiotherapy (2) 3 patients received azithromycin 1, 2, and 3 days before LDRT | 1.5 Gy | - | 12 days | Efficacy endpoints: time to clinical recovery, radiographic improvement, and serologic responses | | Mean time to clinical recovery: 35 h |
| Sharma et al. | June to August 2020 | India | 10 | 51 | Single-fraction whole-lung radiotherapy | 0.7 Gy | No patient required RT interruption due to deterioration of vitals or oxygen saturation | 15 days | Clinical recovery, death, intubation | | Nine patients survived One patient died Clinical recovery: ranging from 3 to 7 days |

**Table A2.** *Cont.*

| Study Author(s) | Start Date | Location | Number of Patients | Mean Age | Interventions | Dose Radiation | Total Oxygen Supplementation Duration | Discharge | Criteria for Efficiency of LDRT | Potential Biases | Outcome |
|---|---|---|---|---|---|---|---|---|---|---|---|
| Sanmamed et al. | April to June 2020 | | 9 | 66 | Single-fraction whole-lung radiotherapy | 1 Gy | Oxygen requirements using $SatO_2/FiO_2$ index (SAFI) at Days 3 and 7 after LDRT | 34 days | Primary outcome: radiological response using severity and extension score on baseline CT at Days 3 and 7 after LDRT Secondary outcomes: toxicity using CTCAE v5, duration of hospitalization, blood work evolution and oxygen requirements using $SatO_2/FiO_2$ index (SAFI) at Days 3 and 7 after LDRT | | Significant changes in the extension score ($p = 0.03$) SAFI index significantly improved 72 h and 1 week after LDRT ($p = 0.01$) Inflammatory blood parameters decreased |
| Papachristofilou et al. | November and December 2020 | University Hospital Basel, Basel, Switzerland | 22 | 75 | Whole-lung low-dose radiation therapy (LDRT) | 1 Gy | - | - | Primary endpoint: ventilator-free days (VFDs) at Day 15 postintervention Secondary endpoints included overall survival, changes in oxygenation, and inflammatory markers | | Whole-lung LDRT failed to improve clinical outcomes in critically ill patients requiring mechanical ventilation for COVID-19 pneumonia |

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
