# Peer review of "Low-Dose Radiation Therapy for COVID-19: A Systematic Review"

_radiation, doi:10.3390/radiation1030020_

Round 1
Reviewer 1 Report
It is acknowledged that the authors greatly improved the manuscript. The remaining issues are presented below.
Old comment: “Inclusion/exclusion criteria are not listed, nor described clearly” - This is an outstanding issue. Although the authors responded that they discuss the criteria in 2.2., the definition and description of what are the criteria is not given (e.g. experimental study, date range, …) – should be corrected.
Old comment: “How can a trial on 1 patient (case study) be included in the SR of clinical trials?”- The justification of including such studies in the review should be given in the manuscript
New comment: sentence “It is well known that X-ray/γ-ray radiotherapy is a cost-effective non toxic cancer treatment that is easily available in most hospitals [44],” is wrong. Radiotherapy is not “non toxic”, often causing side effects in healthy tissue. “non toxic” should be removed.
Reviewer 2 Report
Pages were not numbered as previously noted and authors did not follow this rule to facilitate the reviewer reading
“cost-effective non-toxic cancer treatment»: “non-toxic” is wrong perhaps “low-toxic”
“Linacs are available everywhere»: unfortunately not !!!
“carcenogenisis, cardivascular and CNS damage»: please correct orthograph words – what is relationship between lung irradiation and CNS damage? Is it spinal cord damage? With low dose as less 1.5 Gy?
Figure 4: “between 1 Gy – 1Gy”: 1 Gy is enough
“but also decreases the cancer risk to acceptable level.” Not acceptable – the risk is stochastic, that means without dose threshold
“aged >40 years, and in particular, elderly patients of >60 years of age, regardless of their sex, can provide an acceptable lifetime attributable risks (LAR) of radiation-induced cancer (RIC) for a radiation dose of 0.7 Gy”: very disputable: without cancer and disease, life expectancy duration of these patients is between 30 and 45 years, duration largely long to develop radiation-induced cancers
“Studies conducted on Acute Radiation Sickness (ARS) in Chernobyl shows that an instant dose in the range of 0.5 to 0.7 Gy or higher may cause bone marrow damage »: that does not mean that doses below are not at risk
Results paragraph: please describe more extensively the results, mainly if table, reported them, is in appendix
In total: article should be more objective with less approximations. Results paragraph is not enough informative for readers
Round 2
Reviewer 2 Report
no comment: replies are in accordance with previour reviewer's comments